# Fabrication of Gold Nanostructures on Quartz Crystal Microbalance Surface Using Nanoimprint Lithography for Sensing Applications

**DOI:** 10.3390/mi13091430

**Published:** 2022-08-29

**Authors:** Ryosuke Nishitsuji, Kenji Sueyoshi, Hideaki Hisamoto, Tatsuro Endo

**Affiliations:** 1Department of Applied Chemistry, Graduate School of Engineering, Osaka Metropolitan University, 1-1 Gakuencho, Nakaku, Sakai 599-8531, Osaka, Japan; 2Japan Science and Technology Agency (JST), Precursory Research for Embryonic Science and Technology (PRESTO), 5-3 Yonban-cho, Chiyoda 102-8666, Tokyo, Japan

**Keywords:** quartz crystal microbalance, localized surface plasmon resonance, nanoimprint lithography, sensor

## Abstract

A quartz crystal microbalance (QCM) is a sensor that uses the piezoelectric properties of quartz crystals sandwiched between conductive electrodes. Localized surface plasmon resonance (LSPR) is an analytical technique that uses the collective vibration of free electrons on metal surfaces. These measurements are known as analysis techniques that use metal surfaces and have been applied as biosensors because they allow for the label-free monitoring of biomolecular binding reactions. These measurements can be used in combination to analyze the reactions that occur on metal surfaces because different types of information can be obtained from them. However, as different devices are used for these measurements, the results often contain device-to-device errors and are not accurately evaluated. In this study, we directly fabricated gold nanostructures on the surface of a QCM to create a device that can simultaneously measure the mass and refractive index information of the analyte. In addition, the device could be easily fabricated because nanoimprint lithography was used to fabricate gold nanostructures. As a proof of concept, the nanoparticle adsorption on gold nanostructures was evaluated, and it was observed that mass and refractive index information were successfully obtained without device-to-device errors.

## 1. Introduction

A quartz crystal microbalance (QCM) is a sensor based on the piezoelectric properties of quartz crystals and is used as a mass sensor by sandwiching a thin quartz crystal between two conductive electrodes [1,2]. QCM can measure the amount of adsorption based on the change in resonance frequency when a molecule is adsorbed on the surface, and the following relationship equation is considered for the change in resonance frequency (∆*f_m_*) and the amount of adsorption (∆*m*) [3].
(1)$$\Delta f_{m}=-\frac{2f_{0}^{2}\Delta m}{A_{e}\sqrt{\mu_{q}\rho_{q}}}$$
where *f*_0_ denotes the resonant frequency of the unloaded crystal, *A_e_* is the effective surface area of the crystal, *μ_q_* is the shear modulus of the material of the crystal, and *ρ_q_* is its density. QCM sensors are actively being introduced in various fields, such as chemistry and biology, because they can be used in gas or liquid solvents to perform various measurements [4,5,6]. In particular, the surface adsorption process not only provides information on the amount of adsorption but also on the conformational properties of molecules and the adsorption properties of biomolecules, and it is used to analyze material interactions and surface conditions [7,8].

Surface plasmon resonance (SPR) is a phenomenon induced by resonance when the frequency of collective oscillations of free electrons on the surface of a thin metal film matches the frequency of the incident light [9]. When light is irradiated onto the metallic nanostructure, the SPR remains near the metallic nanostructure and becomes localized. This phenomenon is called localized surface plasmon resonance (LSPR) [10,11]. LSPR is sensitive to changes in the local dielectric environment, such as the refractive index, and can be detected through LSPR wavelength-shift measurements. This principle can be applied to detect the adsorption of chemicals and biomolecules on nanostructured surfaces [12,13]. Therefore, LSPR spectroscopy is a useful technique in chemical and biological sensing and has been applied as a biosensor for antigen–antibody reactions and DNA hybridization [14,15,16].

QCM and LSPR sensors detect the reaction or adsorption occurring at the electrode surface and metal surface, respectively, and are applied as biosensors because they are capable of the label-free detection of biomolecule binding reactions [17,18]. As these measurements can acquire different information, their combination can lead to a detailed analysis of the reaction or adsorption occurring at the metal surface and can be used to analyze the interaction and behavior of biomolecules [19,20,21]. However, because these measurements are made using different devices, the mass and refractive index information obtained by independent measurements are often not identical, and analysis of this information by direct comparison is difficult. A sensor that can measure the mass and refractive index with the same device is required to directly discuss these types of information.

In this study, we directly fabricated gold nanostructures on the electrode surface of a QCM using the nanoimprint lithography (NIL) method, enabling us to measure the mass and refractive index of the adsorbed species with the same device (Figure 1). The NIL method is a nanostructure transfer technology that uses molds with nanostructures, and it is an inexpensive and simple way to fabricate nanostructures [22,23]. In our previous work, we successfully detected local refractive index changes due to DNA hybridization with gold nanostructures fabricated using the NIL method [24]. Therefore, gold nanostructures fabricated by NIL can be used to measure refractive index changes induced by adsorbed substances. In this study, in addition to measuring the change in refractive index due to gold nanostructures, we attempted to simultaneously measure the weight change. Experiments were performed with silica nanoparticles to simultaneously measure the amount of adsorption and the corresponding refractive index when the nanoparticles are adsorbed, as a proof of concept for Figure 1. Using the developed device, we can conclude whether the refractive index changes obtained by the LSPR sensor are due to analyte adsorption/desorption by obtaining mass change information using the same device.

## 2. Materials and Methods

### 2.1. Fabrication of Gold Nanostructure

The gold nanostructures were fabricated using the NIL process (Appendix A). In this study, a cyclo-olefin polymer (COP) film (FLH230/200-120, Scivax Co., Ltd., Kanagawa, Japan) was used as the mold. We selected this structure herein because it has been previously reported by Nishiguchi et al. that plasmon excitation is possible [25]. Initially, the COP mold was cleaned with 2-propanol (Kanto Chemical Co. Inc., Tokyo, Japan) and ultrapure water, and then dried by airflow. A 200-nm-thick Au layer was thermally deposited onto the COP mold. The deposited Au layer was attached to the surface of the QCM (SEN-5P-H-10; TAMADEVICE Co., Ltd., Kanagawa, Japan) using a photocurable polymer (NOA81; Norland Products Inc., Cranbury, NJ, USA), followed by dissolution of the COP mold in D-limonene (NACALAI TESQUE, Inc., Kyoto, Japan), after which a QCM device with a gold nanostructure was obtained.

### 2.2. Observation of Nanostructures

The COP mold and gold nanostructure were observed using field-emission SEM (SU8010; Hitachi, Ibaraki, Japan) at an acceleration voltage of 10 keV. In the observation, Pt was sputtered onto the surfaces with a thickness of approximately 3 nm for clearer observation of the surfaces by SEM.

### 2.3. Optical Characterization of Gold Nanostructure

The reflection spectra of the gold nanostructures were measured using a homemade optical setup (Appendix A). Based on an upright microscope (Wraymer Inc., Osaka, Japan), the setup was composed of a tungsten–halogen lamp, lens, aperture, half-mirror, long-working-distance lens (Sigma Koki Co. Ltd., Hidaka, Japan), spectrometer (USB4000), and operation software (OceanView; Ocean Insight, Orlando, FL, USA). The reflection spectrum of the gold nanostructure on a QCM device was measured under air conditions and compared with the reflection spectrum calculated using a finite-difference time-domain (FDTD) solution (Lumerical Solutions, Inc., Vancouver, BC, Canada). The details of the simulation model are shown in Appendix A. Subsequently, mixed solutions of ultrapure water and 2-propanol (Kanto Chemical Co. Inc., Tokyo, Japan) were prepared, and the refractive indices of the mixed solutions were measured using a refractometer (PAL-RI; ATAGO CO., LTD., Tokyo, Japan) (*n* = 1.3323–1.3752). To measure these solutions, silicone rubber sheets with holes (thickness: 5 mm, f: 10 mm) were placed on the QCM devices with gold nanostructures, and the holes were filled with the mixed solutions. A cover glass was placed on top of the holes to define the liquid thickness. The refractive index sensitivity of the gold nanostructure on the QCM was measured using a homemade optical setup in the same manner as described above.

### 2.4. Measurement of Adsorption Amount and Optical Properties Change

Silica nanoparticle dispersions (1.0 × 10^−4^–3.0 × 10^−2^% (*w*/*w*)) were prepared by dispersing silica nanoparticles (Polysciences, Inc., Warrington, DC, USA) in ultrapure water. Ten microliters of the dispersion were dropped onto the fabricated device, dried at room temperature, and the amount of adsorption was measured using a QCM system (QCM922A; SEIKO EG&G Co., Ltd., Tokyo, Japan). In addition, the reflection spectrum was measured using a custom-built optical setup. The absorption peak shifts for each dispersion concentration were analyzed. In this experiment, nanoparticles were used as analytes because their weight and number could be calculated.

## 3. Results and Discussion

### 3.1. Evaluation of Fabricated Gold Nanostructures

A fabricated QCM device with gold nanostructures is shown in Figure 2a. The red square on the QCM surface represents the transferred region of the gold nanostructure. Red coloration via surface plasmon resonance was confirmed, suggesting that gold nanostructures were successfully fabricated. Figure 2b,c show SEM images of the COP mold and the transferred gold nanostructures, respectively. From the SEM images, the diameters and lattice constants of the holes of the COP mold and pillars of the gold nanostructure were measured. The hole diameter and lattice constant of the COP mold were 211.6 ± 3.9 nm and 430.6 ± 7.6 nm, respectively, while the pillar diameter and lattice constant of the gold nanostructure were 224.9 ± 16.2 nm and 429.6 ± 11.2 nm, respectively. This indicates that the NIL method allowed for the successful transfer of the shape of the COP mold. The increase in the variation of the transferred structures was attributed to the accuracy of the NIL process. Handling, temperature, humidity, and other factors during adhesion to the QCM surface and COP mold removal operations of the fabrication process considerably affected the fabrication accuracy, which has led to variation.

### 3.2. Evaluation of Refractive Index Response of Fabricated Gold Nanostructures

The reflection spectrum was measured in air to confirm the optical properties of the fabricated gold nanostructures. An absorption peak at 660 nm was experimentally observed and evaluated by the theoretical calculation of the reflection spectrum and the electric field distribution based on the FDTD method. Comparing the measured and simulated reflection spectra, the respective absorption peak wavelengths (measured at 660 nm and simulated at 649 nm) were observed to be very close (Figure 3a), suggesting that a structure close to the simulation model was fabricated by the NIL method. However, the measured absorption peak was broader than the calculated absorption peak. We think this is due to slight variations in the fabricated nanostructures. The wavelength of the absorption peak differed slightly from place to place, due to the slight variation in the nanostructure. We speculate that the wavelengths were averaged out in the actual measurement and observed as a broad peak. Additionally, an electric field distribution at 649 nm, calculated by simulation, was formed outside the gold nanostructure (Appendix A). This indicates that the absorption peak is sensitive to the adsorption of substances on the surface of gold nanostructures and can be used to detect adsorbed substances. The refractive index sensitivity of the fabricated gold nanostructures was checked using a mixture of ultrapure water and 2-propanol. It was confirmed that the absorption peak wavelength was red-shifted with an increasing refractive index (Figure 3b,c). This indicates that the gold nanostructures fabricated using NIL have refractive index responsivity, as expected. The shift in the absorption peak wavelength was calculated by simulation (Appendix A) and compared with actual measurements (Figure 3d). The refractive index sensitivities that assume linearity between the peak wavelength shift and refractive index were 233 nm/RIU and 204 nm/RIU for the measured and simulated values, respectively. The experimental refractive-index sensitivity values were similar to those of the simulation. Based on these results, we decided to use the observed absorption peaks for refractive-index sensing.

### 3.3. Correlation between Adsorption and Refractive Index Change

The fabricated device was used to measure the amount of nanoparticle adsorption by dropping the nanoparticle dispersion liquids, followed by drying (Figure 4a). Consequently, a decrease in the resonance frequency was observed as the concentration of the nanoparticle dispersion solution increased. The amount of adsorption was calculated from the measurements of the change in resonance frequency and compared with the results obtained when QCM without a nanostructure was used (Figure 4b). These plots are shown as the average of three measurements, and the error bars represent standard errors. There was no considerable difference in the response between the cases with and without the nanostructure, but the error in the low concentration range was relatively larger in the presence of the nanostructure. In general, as the standard frequency of the QCM increases, the QCM becomes more sensitive and can measure small weight changes, but the upper limit of the measurable weight decreases. In this experiment, a QCM with a low standard frequency (5 MHz) was used to consider the weight of the fabricated gold nanostructures, resulting in a larger error in the low-concentration range. The actual weight of the dropped nanoparticles was calculated to be approximately 10–3000 ng. As the sensitivity of the QCM used in the experiment was approximately 3.47 ng/Hz, the error in the low-concentration range was considered to be larger. A QCM with a higher standard frequency could be used to make the fabricated gold nanostructures lighter. This is expected to allow for a smaller error in the low-concentration range.

The wavelength shift of the absorption peak was also measured after nanoparticle adsorption (Figure 4c,d). From the measurement results, we plotted the relation between the concentration of the nanoparticle dispersion and the absorption peak wavelength shift and confirmed that the absorption peak wavelength was red-shifted as the concentration of the nanoparticle dispersion increased (Figure 4e). This result is the average of three measurements, and the error bars represent the standard errors from the three measurements. The red shift of the absorption peak wavelength in this experiment is attributed to the local refractive index change that occurs when nanoparticles are adsorbed on the gold nanostructure surface. This result led to the successful measurement of the change in optical properties when the measured substance was adsorbed on the surface of the fabricated device. However, the measurement error was also observed to be larger in the low-concentration range. In the structure used in this experiment, the electric field distribution area was relatively small; therefore, the percentage of nanoparticles adsorbed in that area was low at low concentrations. This has led to errors. In this study, a cylindrical pillar structure was used because fabrication using NIL was easy. However, we believe that the detection sensitivity in the low-concentration range can be improved by investigating a structure in which the electric field distribution is more widely formed. In addition, in this experiment, the measurement sensitivity was low because we measured physically adsorbed nanoparticles. The detection sensitivity can be improved using a ligand to immobilize the analytes on the surface. The number of adsorbed nanoparticles was estimated from the measured weight, and the relation between the number and the shift in the refractive index was plotted (Figure 4f). Based on the results shown in Figure 4f, the absorption peak wavelength was red-shifted by 1.53 nm when the number of adsorbed nanoparticles increased by a factor of 10. The fabricated device makes it possible to discuss the correlation between the absolute value of adsorption and refractive index change.

## 4. Conclusions

We successfully fabricated gold nanostructures on QCM devices using the NIL method. The SEM images showed that the hole diameter and lattice constant of the COP mold were comparable to the pillar diameter and lattice constant of the gold nanostructure. This indicated that gold nanostructures reflecting the shape of the COP mold were successfully fabricated using the NIL method. The FDTD method was used to calculate the reflection spectra of the fabricated gold nanostructures. The absorption peak was obtained at a close wavelength, as calculated using simulations. As the electric field distributions at this absorption peak were formed outside the pillar, this absorption peak was responsive to changes in the external environment. The refractive index response was evaluated using a mixture of ultrapure water and 2-propanol, and a red shift of the absorption peak wavelength was observed as the peripheral refractive index increased. The refractive-index sensitivity was also compared with the simulation results, and comparable values were obtained. Finally, the fabricated devices were evaluated by a model experiment using nanoparticles, and the changes in the resonance frequency and optical properties were successfully measured when nanoparticles were adsorbed on the device surface. These results suggest that the same device can successfully measure optical property changes based on LSPR and the amount of analyte adsorbed on the device surface based on QCM, and that this device can evaluate mass and refractive index information simultaneously without device-to-device errors.

## Figures and Tables

**Figure 1 micromachines-13-01430-f001:**
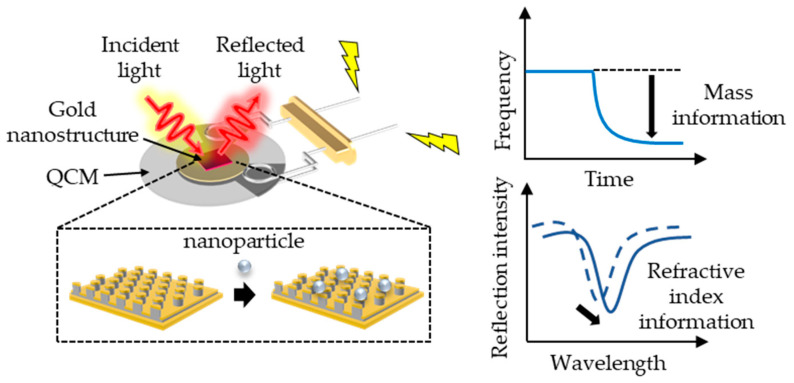
Concept of a device for simultaneously measuring mass and corresponding refractive index by QCM and LSPR sensor.

**Figure 2 micromachines-13-01430-f002:**
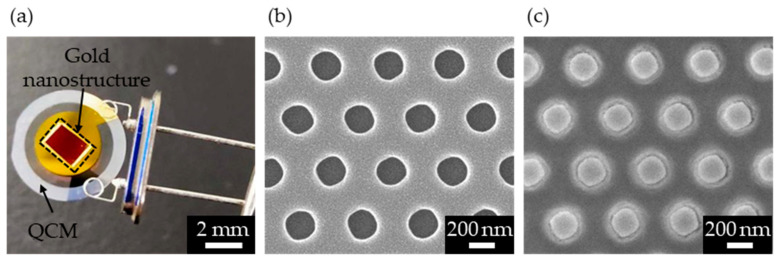
(**a**) Image of the fabricated QCM device with gold nanostructures. (**b**) SEM image of the COP mold (hole). (**c**) SEM image of the gold nanostructure (pillar).

**Figure 3 micromachines-13-01430-f003:**
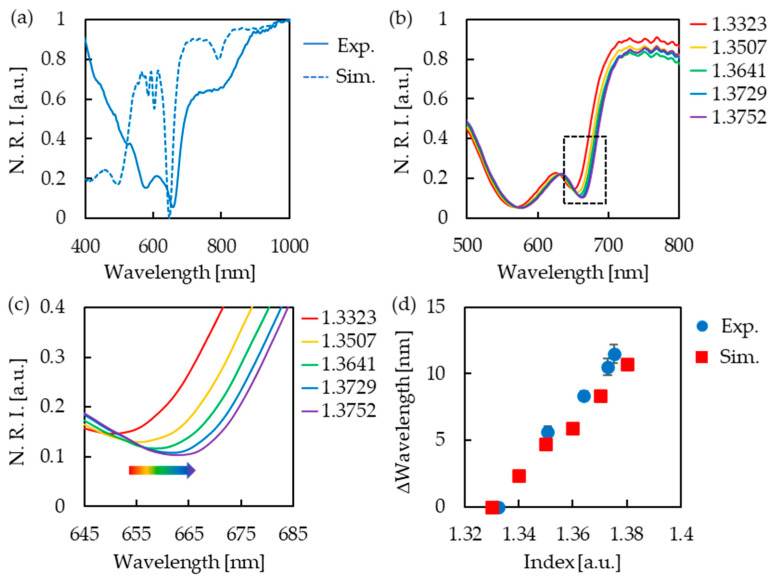
(**a**) Reflection spectra of experiments (solid lines) and simulations (dashed lines). The vertical axis shows normalized reflection intensity (N. R. I.). (**b**) Reflection spectra of gold nanostructures in various refractive index solutions (water/2-propanol mixture). (**c**) Enlarged figure of absorption peak in (**b**) (dotted square part). The peak wavelength was red-shifted with the increasing refractive index. (**d**) Comparison of experimental and simulated peak shift.

**Figure 4 micromachines-13-01430-f004:**
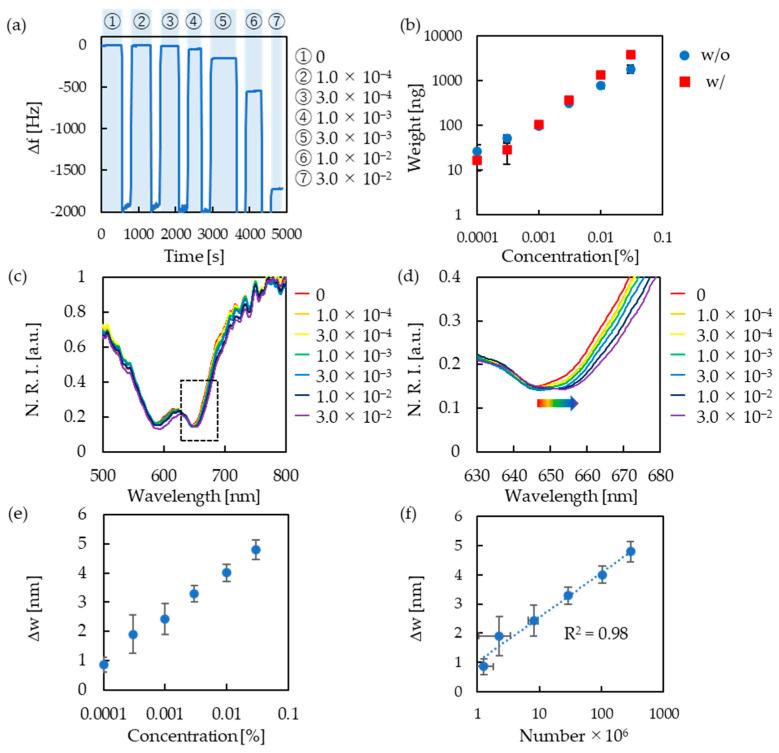
(**a**) Resonance frequency change measured by a QCM devices with nanostructure when nanoparticles (1.0 × 10^−4^–3.0 × 10^−2^% (*w*/*w*)) are adsorbed. The vertical axis shows ∆frequency (∆f). (**b**) Comparison of weight response to nanoparticle concentrations with and without a nanostructure on a QCM device. (**c**) Reflection spectra of gold nanostructure in various concentrations of nanoparticle dispersion (1.0 × 10^−4^–3.0 × 10^−2^% (*w*/*w*)). (**d**) Enlarged figure of absorption peak in (**c**) (dotted square part). (**e**) Peak wavelength shift with different concentrations of nanoparticle dispersion. The vertical axis shows ∆wavelength (∆w). (**f**) Peak wavelength shift with number of adsorbed nanoparticles.

## Data Availability

The data are contained within the article and Appendix A.

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
