# Peer review of "Fabrication of Gold Nanostructures on Quartz Crystal Microbalance Surface Using Nanoimprint Lithography for Sensing Applications"

_micromachines, 2022, doi:10.3390/mi13091430_

Round 1
Reviewer 1 Report
The authors reported the NIL method to fabricate gold nanostructures on QCM devices. The English writing of the manuscript is good. Some details and contents need to be added in this manuscript carefully before publication:
1. The type of the formula (1) is not right, please change it to a normal type.
2. According to the Sauerbrey theory, the premise of the application of the Sauerbrey equation is that the attachment must be uniformly and rigidly adsorbed on the electrode surface. However, the Figure 2 shows that the distribution of the nanostructures on the electrode surface is not uniform. How can you use the Sauerbrey equation?
3. In Figure 3, why don’t the experiments (solid lines) and simulations (dashed lines) match?
4. Please give the advantages of adding nanostructures on the QCM electrode surface.
5. Please add the experiments about the application of the QCM electrode surface with nanostructures, and compare the advantages of QCM with/without nanostructures.
Author Response
Reviewer 1
The authors reported the NIL method to fabricate gold nanostructures on QCM devices. The English writing of the manuscript is good. Some details and contents need to be added in this manuscript carefully before publication:
Thank you very much for your comments and advice. Our comments to your revisions are following.
- The type of the formula (1) is not right, please change it to a normal type.
Response: Thank you for your comment. Formula (1) has been modified.
- According to the Sauerbrey theory, the premise of the application of the Sauerbrey equation is that the attachment must be uniformly and rigidly adsorbed on the electrode surface. However, the Figure 2 shows that the distribution of the nanostructures on the electrode surface is not uniform. How can you use the Sauerbrey equation?
Response: Thank you for your comment. In this paper, gold nanostructure was fabricated on a part of the QCM electrode and silica nanoparticles were dropped onto the gold nanostructure. As you point out, it does not strictly follow Sauerbrey equation. However, we had confirmed that the weight of silica nanoparticles calculated from the drop volume and concentration of the nanoparticle dispersion matched well with the weight of silica nanoparticles calculated from QCM measurements, so we used Sauerbrey equation as an approximate formula.
- In Figure 3, why don’t the experiments (solid lines) and simulations (dashed lines) match?
Response: Thank you for your comment. The simulation model was an endless series of exactly the same structures. On the other hand, the structures used in the experiments were fabricated by nanoimprint lithography and were slightly different in parts. This difference is the reason why experiments (solid lines) and simulations (dashed lines) do not completely match.
- Please give the advantages of adding nanostructures on the QCM electrode surface.
Response: Thank you for your comment. As QCM and LSPR sensors can acquire different information, their combination can lead to a detailed analysis of the reaction or adsorption occurring at the metal surface and can be used to analyze the interaction and behavior of biomolecules. However, because these measurements are made using different devices, the mass and refractive index information obtained by independent measurements are often not identical, and analysis of this information by direct comparison is difficult. Since a QCM with nanostructure allows mass and refractive index to be measured on the same device, this information can be discussed directly.
- Please add the experiments about the application of the QCM electrode surface with nanostructures, and compare the advantages of QCM with/without nanostructures.
Response: Thank you for your valuable advice. All data in Figure 4 were acquired using QCM with nanostructure. The advantage of using QCM with nanostructure is that the mass and refractive index information can be obtained with a same device. In addition, there was no considerable difference in the response of QCM between the cases with and without the nanostructure.

Reviewer 2 Report
Title: Fabrication of gold nanostructures on QCM surface using nanoimprint lithography for sensing applications
This paper focuses on developing a single platform for conducting LSPR based sensing by fabricating gold nanoparticles on a QCM surface to reduce the device-to-device error involved in LSPR sensing. The developed platform was validated by sensing the mass and refractive index of the gold nanoparticles.
The authors begin by explaining how they coated the QCM with gold nanoparticles using NIL technique and validated it through SEM imaging. This is followed by a discussion regarding comparison of the experimental and simulation data for NRI. The authors also studied the NRI of gold shift under presence of other solvents. Finally, they discuss how the peak wavelength shift, NRI and frequency change correlate with the number of nanoparticles deposited on the QCM surface.
Overall, the reported data in the article seems promising (although obvious), but I believe that it lacks a lot of key fundamental discussions. Figures are also poorly presented, and some terminologies are not well explained. Hence, I do not recommend the publication of this article at its current state. Here are my comments:
1. Why did the author choose gold for deposition?
2. What happens to the signal change when the molecule size is large, eg: protein?
3. How are the authors being sure that the signal is not being disturbed by external conditions? Since they are dealing with nanoparticles, it is obvious external conditions might affect the sensing. Clarification needed.
4. Is the process reversible? Can you remove all the deposited nanoparticles and re-deposit again to check the sensitivity?
5. How does the signal change if the rate of deposition changes with time?
6. If the authors are talking about label-free sensing, why don’t they show a proof-of-concept with an analyte?
Author Response
Reviewer 2
This paper focuses on developing a single platform for conducting LSPR based sensing by fabricating gold nanoparticles on a QCM surface to reduce the device-to-device error involved in LSPR sensing. The developed platform was validated by sensing the mass and refractive index of the gold nanoparticles.
The authors begin by explaining how they coated the QCM with gold nanoparticles using NIL technique and validated it through SEM imaging. This is followed by a discussion regarding comparison of the experimental and simulation data for NRI. The authors also studied the NRI of gold shift under presence of other solvents. Finally, they discuss how the peak wavelength shift, NRI and frequency change correlate with the number of nanoparticles deposited on the QCM surface.
Overall, the reported data in the article seems promising (although obvious), but I believe that it lacks a lot of key fundamental discussions. Figures are also poorly presented, and some terminologies are not well explained. Hence, I do not recommend the publication of this article at its current state. Here are my comments:
Response: Thank you for your comment. We have modified the figures and added explanations.
Thank you very much for your comments and advice. Our comments to your revisions are following. In addition, we have revised our manuscript as you have indicated.
- Why did the author choose gold for deposition?
Response: Thank you for your comment. This is because gold is a widely used material in the LSPR. It is also a chemically stable and biocompatible material.
- What happens to the signal change when the molecule size is large, eg: protein?
Response: Thank you for your comment. Both the refractive index change and the change in adsorption are expected to be larger, making their measurement easier.
- How are the authors being sure that the signal is not being disturbed by external conditions? Since they are dealing with nanoparticles, it is obvious external conditions might affect the sensing. Clarification needed.
Response: Thank you for your comment. We think that external factors are present in the measurement. However, in this experiment, we observed changes before and after the adsorption of analytes, so we expect that external factors canceled out and the effect became smaller.
- Is the process reversible? Can you remove all the deposited nanoparticles and re-deposit again to check the sensitivity?
Response: Thank you for your comment. It can be used repeatedly if adsorbed substances can be removed. However, since the silica nanoparticles used in this experiment could not be completely removed, reusability has not been investigated.
- How does the signal change if the rate of deposition changes with time?
Response: Thank you for your comment. In this experiment, nanoparticles are physically adsorbed on the electrode surface by dropping and drying the nanoparticle dispersion. Therefore, the experimental results in this paper do not allow for a discussion of adsorption rates. Assuming measurements in solution, the faster the adsorption rate, the greater the amount of change per unit time, but there is no difference in the final amount of change.
- If the authors are talking about label-free sensing, why don’t they show a proof-of-concept with an analyte?
Response: Thank you for your comment. In this experiment, nanoparticles were used because their weight and number can be calculated and it is easy to evaluate the performance of the fabricated devices. Label-free sensing can be achieved by measuring antigen-antibody reactions with this device. Hence, in the future, we will try to detect the label-free detection of antigen-antibody reactions.

Reviewer 3 Report
Authors of the manuscript titled "Fabrication of gold nanostructures on QCM surface using nanoimprint lithography for sensing applications "made a device capable of simultaneously measuring the analyte's mass and refractive index information and fabricated gold nanostructure using nanoimprint lithography on the surface of Quartz crystal microbalance (QCM) to do so. The topic is within the scope of the journal. However, the manuscript needs major and sincere revision before accepting:
1) The abbreviation "QCM" on the manuscript title appeared for the first time within the manuscript. This is a little confusing at first glance. It is suggested to use the full form of it.
2) Authors should mention somewhere in the manuscript, especially in the material section, that they used gold as the analyte. Newbie readers would find it hard to understand as many analytes are out there like 24 karat gold, NaCl, water, etc.
3) The overall manuscript requires an English language check. Improper preposition was used in many places, like line 15 "occurring at" should be "occurring on," singular verb and plural verb should be checked again, such as line 22 "gold nanostructures was evaluated" here was should be 'were'. Then again, "is used', "was used' is confusing. It is suggested to be consistent about the tense used.
4) Can the author write few key factors and data's from their previous research work red 24 so that the readers can easily relate to the work has been done here and comprehend the proof of concept. This manuscript should carry out its own significant. Therefore line 69-74 is a little confusing at this moment.
5) Line 73 'adsorption/desorption or simple structural changes', Line 74' adsorption/desorption' has made the whole statement of 71-75 confusing.
6) How come 2 Materials and Methods and 3 Result and Discussion have exactly the same point names. This alone makes the overall manuscript insignificant. The authors should consider drafting the manuscript again in a systematic manner. Keeping the methods to achieve the device and the materials used to do so as a sole concern of point 2 maybe a good idea. Then again the point 3 may contain the final outcome and discussing how the outcome came as well as relating them with the point 2 would enrich the manuscript.
7) Line 88, because it has been reported …… this statement is poorly written. It should be like because it has been previously reported by name et al. …... in this manner or so. This is easier to comprehend for the reader from where information came from along with the common referencing technique.
8) 145-147' other factors' could be anything, can author say what kind of other factor ? What the authors did to get the optimum accuracy of the fabrication process.
9) Line 223, 147, 160, 219, 204; the use 'may' like may have, may be due to …. This are very vague and unclear. It seems like the authors don't know so they want to say either this or this. This makes the statements not scientifically proven. This kind of errors needs a through fixing in the overall manuscript.
10) How the authors calculated the error mentioned in line 216.
11) Line 182 what the authors mean by red-shifted?
12) The claim not having device-to-device error observation have no follow-up result or visual comparison. The claim is falsified. The whole significance of the manuscript is this. Which is totally ignored. In the discussion section this could be discussed as a separate point. Having previous data with the new data from where they proved that or claimed that they made a device-to-device error-free observation.
A graphical figure to illustrate the overall process is another good way to make the work more understandable and attractive. It should be kept in mind, the manuscript is for the readers and way to communicate with the community with the scientific discoveries made, should not be self-explanatory.
Author Response
Reviewer 3
Authors of the manuscript titled "Fabrication of gold nanostructures on QCM surface using nanoimprint lithography for sensing applications "made a device capable of simultaneously measuring the analyte's mass and refractive index information and fabricated gold nanostructure using nanoimprint lithography on the surface of Quartz crystal microbalance (QCM) to do so. The topic is within the scope of the journal. However, the manuscript needs major and sincere revision before accepting:
Thank you very much for your comments and advice. Our comments to your revisions are following.
- The abbreviation "QCM" on the manuscript title appeared for the first time within the manuscript. This is a little confusing at first glance. It is suggested to use the full form of it.
Response: Thank you for your valuable advice. The title was revised (QCM→quartz crystal microbalance).
- Authors should mention somewhere in the manuscript, especially in the material section, that they used gold as the analyte. Newbie readers would find it hard to understand as many analytes are out there like 24 karat gold, NaCl, water, etc.
Response: Thank you for your valuable advice. In this experiment, silica nanoparticles were used as the analyte; this has been added in Materials and Methods 2.4. Gold is used for nanostructure fabrication.
- The overall manuscript requires an English language check. Improper preposition was used in many places, like line 15 "occurring at" should be "occurring on," singular verb and plural verb should be checked again, such as line 22 "gold nanostructures was evaluated" here was should be 'were'. Then again, "is used', "was used' is confusing. It is suggested to be consistent about the tense used.
Response: Thank you for your valuable advice. The overall manuscript was checked and corrected by the native speaker.
- Can the author write few key factors and data's from their previous research work red 24 so that the readers can easily relate to the work has been done here and comprehend the proof of concept. This manuscript should carry out its own significant. Therefore line 69-74 is a little confusing at this moment.
Response: Thank you for your valuable advice. We have added a detailed description.
- Line 73 'adsorption/desorption or simple structural changes', Line 74' adsorption/desorption' has made the whole statement of 71-75 confusing.
Response: Thank you for your valuable advice. We have simplified the statement.
- How come 2 Materials and Methods and 3 Result and Discussion have exactly the same point names. This alone makes the overall manuscript insignificant. The authors should consider drafting the manuscript again in a systematic manner. Keeping the methods to achieve the device and the materials used to do so as a sole concern of point 2 maybe a good idea. Then again the point 3 may contain the final outcome and discussing how the outcome came as well as relating them with the point 2 would enrich the manuscript.
Response: Thank you for your valuable advice. The manuscript was modified.
- Line 88, because it has been reported …… this statement is poorly written. It should be like because it has been previously reported by name et al. …... in this manner or so. This is easier to comprehend for the reader from where information came from along with the common referencing technique.
Response: Thank you for your valuable advice. The statement was modified.
- 145-147' other factors' could be anything, can author say what kind of other factor ? What the authors did to get the optimum accuracy of the fabrication process.
Response: Thank you for your comment. We think that the difference in the surface condition of the mold and the mold cleaning operation are also the cause. Optimization was performed for the amount of photocurable polymer drops during adhesion to a QCM electrode surface and the temperature of the COP mold removal process.
- Line 223, 147, 160, 219, 204; the use 'may' like may have, may be due to …. This are very vague and unclear. It seems like the authors don't know so they want to say either this or this. This makes the statements not scientifically proven. This kind of errors needs a through fixing in the overall manuscript.
Response: Thank you for your valuable advice. The indicated points have been modified.
- How the authors calculated the error mentioned in line 216.
Response: Thank you for your comment. The error bars represent the standard errors from three measurements.
- Line 182 what the authors mean by red-shifted?
Response: Thank you for your comment. It means that the peak wavelength was shifted to the longer wavelength side. Figure 3c was modified because the explanation was difficult to understand.
- The claim not having device-to-device error observation have no follow-up result or visual comparison. The claim is falsified. The whole significance of the manuscript is this. Which is totally ignored. In the discussion section this could be discussed as a separate point. Having previous data with the new data from where they proved that or claimed that they made a device-to-device error-free observation.
Response: Thank you for your comment. The measurement of adsorption by QCM and the measurement of optical property changes by LSPR usually need to be performed on different devices, so inter-device errors exist in these measurements. In this experiment, these measurements were performed on the same device by using a QCM with gold nanostructures. Therefore, these measurements have no errors due to differences in devices.
A graphical figure to illustrate the overall process is another good way to make the work more understandable and attractive. It should be kept in mind, the manuscript is for the readers and way to communicate with the community with the scientific discoveries made, should not be self-explanatory.
Response: Thank you for your valuable advice. I have revised the sections you pointed out.

Round 2
Reviewer 1 Report
The authors have modified the paper well. However, I have a suggestion to author below. In Figure 4 (a), (e) and (f), the mark of the ordinate should be changed to ∆f, ∆w and ∆w.
Reviewer 2 Report
Accept the article in its present form
Reviewer 3 Report
The manuscript titled Fabrication of gold nanostructures on quartz crystal microbalance surface using nanoimprint lithography for sensing applications is within the journal's scope. The manuscript is recommended for publication.